# Serological Evidence That SARS-CoV-2 Has Not Emerged in Deer in Germany or Austria during the COVID-19 Pandemic

**DOI:** 10.3390/microorganisms10040748

**Published:** 2022-03-30

**Authors:** Andres Moreira-Soto, Christian Walzer, Gábor Á. Czirják, Martin H. Richter, Stephen F. Marino, Annika Posautz, Pau De Yebra Rodo, Gayle K. McEwen, Jan Felix Drexler, Alex D. Greenwood

**Affiliations:** 1Institute of Virology, Charité-Universitätsmedizin Berlin, Humboldt-Universität zu Berlin, Berlin Institute of Health, 10117 Berlin, Germany; andres.moreira-soto@charite.de; 2Wildlife Conservation Society, Bronx, NY 10460, USA; cwalzer@wcs.org; 3Research Institute of Wildlife Ecology (FIWI), A-1160 Vienna, Austria; annika.posautz@vetmeduni.ac.at; 4Department of Wildlife Diseases, Leibniz Institute for Zoo and Wildlife Research, 10315 Berlin, Germany; czirjak@izw-berlin.de (G.Á.C.); rodo@izw-berlin.de (P.D.Y.R.); mcewen@izw-berlin.de (G.K.M.); 5Bundesinstitut für Risikobewertung (BfR), 10589 Berlin, Germany; martin.richter@bfr.bund.de (M.H.R.); stephen-francis.marino@bfr.bund.de (S.F.M.); 6Department of Veterinary Medicine, Freie Universität Berlin, 14163 Berlin, Germany

**Keywords:** COVID-19, SARS-CoV-2, roe deer, red deer, fallow deer, hunting

## Abstract

Spillover of severe acute respiratory syndrome coronavirus type 2 (SARS-CoV-2) to North American white-tailed deer (*Odocoileus virginianus*) has been documented. However, it is unclear if this is a phenomenon specific to North American deer or is a broader problem. We evaluated pre and pandemic exposure of German and Austrian deer species using a SARS-CoV-2 pseudoneutralization assay. In stark contrast to North American white-tailed deer, we found no evidence of SARS-CoV-2 exposure.

## 1. Introduction

The COVID-19 pandemic caused by the SARS-CoV-2 virus has spread from humans to domestic animals, captive animals, and free ranging wildlife [1,2,3,4,5,6]. In some cases, the result was the spillback of a novel SARS-CoV-2 variant to humans, making new wildlife reservoirs a clear threat to human health. Spillover is of particular concern in North America where there is strong evidence that white-tailed deer (*Odocoileus virginianus*) have been infected in the U.S. and Canada with the prevalence reaching 82.5% [3]. White-tailed deer may now represent a reservoir in which SARS-CoV-2 may evolve and potentially spill back to humans with unpredictable health consequences. White-tailed deer are social animals that often form large herds, are important game species, and often live in peri-urban and urban environments, making them unsurprising candidates as a viral reservoir species for pathogens with zoonotic potential. However, it is unclear how generalized spillback is in deer. In Europe, multiple species of deer are heavily managed and hunted but hunting legislation and management practices differ substantially between the U.S. and Europe, within Europe, and even within individual countries and regions. Different deer species may also vary in the sequence of their angiotensin converting enzyme-2 (ACE2) receptor (the receptor with which the SARS-CoV-2 virus spike protein interacts), which could hinder or promote infection [7].

To assess whether SARS-CoV-2 spillover has occurred in Europe, we applied a SARS-CoV-2 pseudoneutralization assay to pandemic-collected sera from the three main species of European cervids, namely red deer (*Cervus elaphus*), roe deer (*Capreolus capreolus*), and fallow deer (*Dama dama*). Samples were collected from Germany and Austria, two countries with long deer management traditions. We also compared the ACE2 receptor sequences for all deer and related bovids in the databases including roe and red deer to determine if they have amino acid changes that would be consistent with resistance to infection with SARS-CoV-2. 

## 2. Materials and Methods

Sera were collected from red deer (*Cervus elaphus*) (n = 67), roe deer (*Capreolus capreolus*) (n = 97), and fallow deer (*Dama dama*) (n = 68) between January 2020 and December 2021 (Table 1 and Appendix A). 

A SARS-CoV-2 surrogate virus neutralization test (sVNT; GenScript, Piscataway, NJ, USA), which measures the antibody-mediated inhibition of SARS-CoV-2 receptor-binding domain (RBD)-ACE2 interaction, was used to test the samples [7]. Briefly, equal volume of deer sera was pre-incubated with HRP-conjugated RBD for 30 min at 37 °C, followed by adding it to an ACE2-coated ELISA plate for 15 min at room temperature. A colorimetric signal was developed using TMB substrate and stopped with 1 M HCl. Absorbance values at 450 nm were acquired using a microplate reader. This test achieves 99.93% specificity and 95–100% sensitivity and the species-independent performance has been validated in rabbit, ferret, mouse, cat, dog, hamster, and alpaca serums [8,9]. The same assay was employed in North American white-tailed deer and by the U.S. in Brazilian cats with high seroprevalence [2,10], corroborating the robustness of this testing.

Sample size calculations required a minimum of 81–138 animals, assuming seroprevalences ranged from 30% to 90%, there was a large population size, and 95% confidence intervals (calculations were performed using the package epiDisplay version 3.5.0.1 in R version R-4.1.3 (.r-project.org/). To assess assay specificity, we further tested pre-pandemic sera from Germany (n = 201) (Table 1). The assay cutoff for the cPass SARS-CoV-2 Neutralization Antibody Detection Kit is 30% inhibition (Figure 1).

A full multiple-sequence alignment of the ACE2 protein for the species is characterized in [7], and additional deer species are included from GenBank (GCA_000751575.1, XP_037678579.1, XP_012949915.3, QEQ50331.1, KAB0345583.1, XP_043752042.1, and XP_036696353.1). Alignments were performed using ClustalW Multiple Alignment in BioEdit version 7.2.5 (Figure 2 and Appendix A).

## 3. Results

All positive and negative controls were in the 450 nm range stated in the manufacturer’s instructions. In stark contrast to North American white-tailed deer, no pre- or pandemic deer sera collected tested positive for SARS-CoV-2 antibodies (Figure 1, Table 1 and Appendix A). None of the samples approched the assay cutoff.

Alignment of the ACE2 receptor among deer and related bovids showed that apart from K-N substitution in the red deer relative to white-tailed deer, none of the sequenced deer species had any substitutions that distinguished them from SARS-CoV-2-susceptible cervid species or other susceptible mammals (Figure 2 and Appendix A).

## 4. Discussion

The contrast between our results and the multiple reports of high prevalence infection in North American white-tailed deer may have several explanations. First, a single relevant amino acid change was observed in red deer at position 31 (K-N) in the ACE2 receptor compared to other deer and bovids. While this could, in principle, explain infection resistance in red deer, roe deer were identical at all sites of import to white-tailed deer, yet were not infected in our study. It is not clear if this difference would make red deer resistant as other susceptible species vary at this position [7] (Figure 2). For sample size calculations, we assumed equal seroprevalences in all deer species. Since there is no preliminary data on susceptibility to SARS-CoV-2 in European deer species, aligment of ACE2 receptor residues amongst cervids suggest similar susceptibility as white-tailed deer, and all European deer share comparable habitats leading to probable similar contact with SARS-CoV-2. In addition, approximately 40% of tested white-tailed deer in the U.S. were positive using the same test [2]. Thus, even if infection occurs at a very low prevalence, the differences between North America and central Europe remain striking.

In Europe, the historic feudal systems, land use, and land allocation practices shaped hunting traditions. Across Germany, Austria and a large part of Switzerland, hunting is implemented using a district-based hunting system (Revier) and wildlife belongs to no one (*res nullius*) [11]. In contrast, in North America, wildlife is considered a public resource managed by the government regardless of the land or water where wildlife lives and is tightly linked to wildlife conservation (*res communis*). This distinction is relevant to the extent to which government bodies can control and influence the management of wildlife and hunting practices. Therefore, in Europe, deer management is more local than in North America [12].

Across the U.S. and Canada, the deer hunting season is generally short, lasting three to four months, while in Europe it varies widely, from less than four weeks in some areas of Switzerland to 213–300 or more days in Austria and Germany [13]. Hunting practices vary across the two regions; in North America, deer hunting over bait and attractants is allowed in over 20 states, while in many regions of Europe, this practice is either prohibited or frowned upon [1]. Across North America, feeding practices range widely by region; Permittances range from allowed to strongly discouraged to illegal. In continental Europe, supplementary seasonal feeding of deer is a common practice associated with, among other things, maintaining high densities and decreasing damage to agriculture and forestry assets [11,14]. However, the “Revier” structure maintains these concentrations locally and in rural areas. A major difference between white-tailed deer and red deer regarding SARS-CoV-2 spillback from humans is that the former thrives seasonally at high densities in urban and peri-urban landscapes [15].

A combination of extensive versus limited feeding, which could expose animals to human exudates (including aeorsols) in contaminated feed, and high density in urban and peri-urban landscapes that could expose herds to human waste, may be the key difference in SARS-CoV-2 spillback to deer populations. The lack of urban or peri-urban populations and the Revier management system likely limits contact among central European deer populations and may diminish human–wildlife interactions sufficiently to prevent the spread of infection. In this context, it would be advisable in Europe to maintain and strengthen barriers to spread, for example, to ensure that deer kept in wildlife parks have no contact with free-ranging deer, and to prevent deer from settling near towns and villages to minimize future spillovers. In North America, it appears important to address behaviors related to feeding of deer in urban settings and to revisit feeding practices in deer management and hunting. Maintaining natural barriers will help prevent future spillovers and spillback of SARS-CoV-2 between deer and human populations [16]. The consequences of inaction could be further spillback to humans of viral variants evolving in novel hosts, as has occurred in Danish and Dutch mink farms during the pandemic [4,5].

## Figures and Tables

**Figure 1 microorganisms-10-00748-f001:**
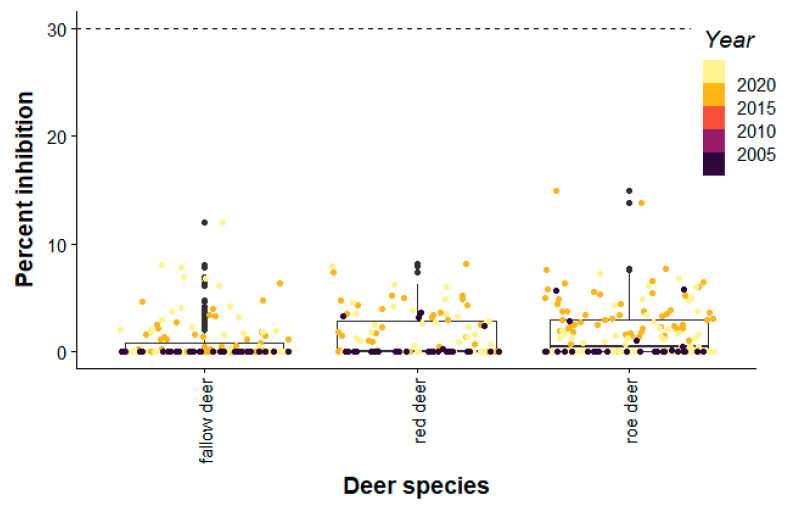
Results of the pseudoneutralization point assay are shown by year and deer species. Horizontal dashed line indicates the assay cutoff of the cPass SARS-CoV-2 Neutralization Antibody Detection Kit as 30% inhibition.

**Figure 2 microorganisms-10-00748-f002:**
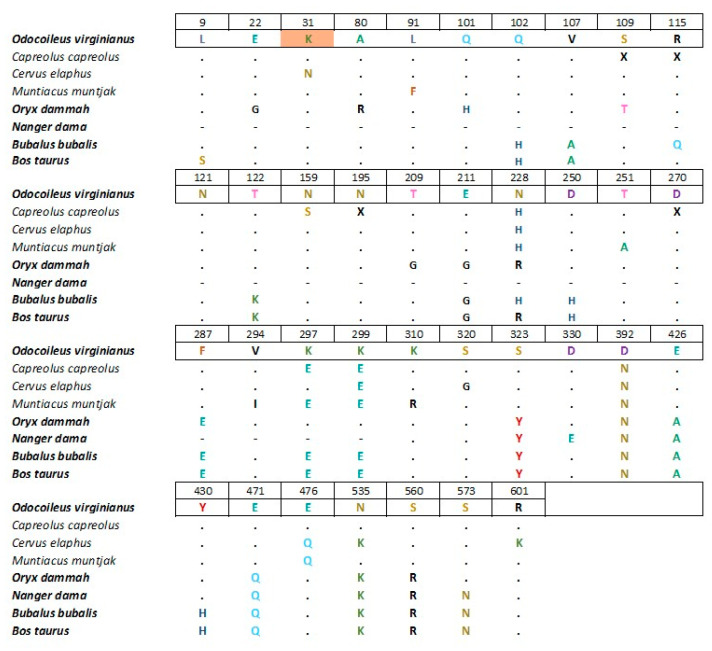
Multiple sequences the ACE2 protein were performed with the species characterized in [7], and additional deer species included from GenBank (GCA_000751575.1, XP_037678579.1, XP_012949915.3, QEQ50331.1, KAB0345583.1, XP_043752042.1, XP_036696353.1). The full alignment is in Appendix A. The figure only shows deer and related bovids and only positions where there are amino acid changes relative to white-tailed deer (*Odocoileus virginianus*). Alignments were performed using ClustalW Multiple Alignment in BioEdit version 7.2.5. The names of species highlighted in bold correspond to species known to be susceptible to SARS-CoV-2 and the amino acid position highlighted in orange is a hot spot for SARS-CoV-2 S-binding [7]. Ambiguities in database sequences are designated as X. Dots represent identity to the reference and dashes represent missing sequence information.

**Table 1 microorganisms-10-00748-t001:** Number and origin of samples from each deer species analyzed for anti- SARS-CoV-2 antibodies.

	Germany		Austria
	Pre-Pandemic	Pandemic	Pandemic
Roe deer	71	97	
Red deer	45	16	51
Fallow deer	85	68	
Total	201	181	51

## Data Availability

Not applicable.

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
