# Peer review of "Serological Evidence That SARS-CoV-2 Has Not Emerged in Deer in Germany or Austria during the COVID-19 Pandemic"

_microorganisms, 2022, doi:10.3390/microorganisms10040748_

Round 1
Reviewer 1 Report
I appreciate the opportunity to review the manuscript, Serological Evidence that SARS-CoV-2 Has Not Emerged in Deer in Germany or Austria during the COVID 19 Pandemic by Moreira-Soto etal. The brief report is well and clearly written and the results are important to disseminate. I only have very minor comments.
--Line 13: white-tailed deer generally includes a hyphen between the “white” and the “tailed.” If updated, change throughout the ms.
--134: The authors might consider aerosol transmission as a potential transmission route in this discussion.
Line 139: it appears as though there is a missing word in this sentence? Possibly need to add an “and”:
…populations “and” may diminish human-wildlife interactions sufficiently to prevent the spread…
Author Response
I appreciate the opportunity to review the manuscript, Serological Evidence that SARS-CoV-2 Has Not Emerged in Deer in Germany or Austria during the COVID 19 Pandemic by Moreira-Soto etal. The brief report is well and clearly written and the results are important to disseminate. I only have very minor comments.
Response:
We thank the reviewer for their positive assessment of our manuscript and helpful comments.
--Line 13: white-tailed deer generally includes a hyphen between the “white” and the “tailed.” If updated, change throughout the ms.
Response: We thank the reviewer for noticing this error. We have fixed it throughout.
--134: The authors might consider aerosol transmission as a potential transmission route in this discussion.
Response: Aerosols are mentioned now as a potential contaminating human exudate.
Line 139: it appears as though there is a missing word in this sentence? Possibly need to add an “and”:
…populations “and” may diminish human-wildlife interactions sufficiently to prevent the spread…
Response: We thank the reviewer for noticing this error and have added the word “and”
Reviewer 2 Report
The manuscript is well written and the methods have been described very well.
It can be accepted as is.
Author Response
Reviewer 2
The manuscript is well written and the methods have been described very well.
It can be accepted as is.
Response: We thank the reviewer for their positive assessment of our manuscript
Reviewer 3 Report
There is a great deal of interest in reverse zoonotic transmission of SARS-CoV-2 and the manuscript by Moreira-Soto and colleagues provides additional information about the force of infection in wild animals in a COVID-endemic environment. The authors performed a serological screening on pre-pandemic and pandemic sera of 3 different deer species, using a previously validated pseudoneutralization assay method (Chandler et al), which allowed to confirm no prevalence of infection in Germany and Austria. Moreover, the authors performed an “in-silico” analysis to compare ACE2 receptor sequences in different animal species.
Although the aim of the work is overall clear, the text needs some minor language revision in order to make the reading of the text smoother, together with a better organization of presented data.
The sample size used by the authors is sufficient and adequate to infer results on the prevalence of infection on 3 different deer species although a main concern can be recognized in the lack of information about the geographical provenance of collected sera and the lockdown measures eventually adopted in those territories. Since Germany and Austria have an important deer population that is extending throughout the nations better geographical identification is mandatory, also considering that the 230 animals tested are a low percentage of the total deer population considered in comparison with the total numbers.
Here are some minor comments.
Line 38 pseudoneutralization assay instead of “assays”
Line 46 “Sera was”.. sera were
Line 53 “equal volume of deer sera was pre-incubated ” instead of were
Line 56 “Absorbance values at 450 nm were” instead of readings
Line 64 “All positive and negative controls were in the 450 nm range stated in the manufacturer’s instructions.” I suggest moving this statement in the results section
Figure 1 caption. “All positive and negative 69 controls were in the 450 nm range stated in the manufacturer’s instructions.” How can be distinguished from other samples? Are there in the image? Otherwise let this statement only in the M&M section. As well as “None of the samples approached the cutoff” can be written only in the result section.
“Results of the Pseudoneutralization point assay are shown by year and deer species. All positive and negative controls were in the 450 nm range stated in the manufacturer’s instructions. The assay cutoff for the cPass SARS-CoV-2 Neutralization Antibody Detection Kit is 30% inhibition which is is indicated. None of the samples approched the cutoff.”
It should be read “ Results of the Pseudoneutralization point assay are shown by year and deer species. Horizontal dashed line indicates the assay cutoff of the cPass SARS-CoV-2 Neutralization Antibody Detection Kit as 30% inhibition.”
Line 73 Check reference number. I suppose the reference the authors are referring to is the number 6 instead the number 5. The same for line 79, 85, and Supplementary figure caption.
Figure 2 caption. “Multiple sequences the ACE2 protein were performed with the species characterized in (5) and additional deer species included from GenBank. . The full alignment is in Supplementary Figure S1” It should be read “ Multiple sequence alignment of ACE2 protein of species characterized in (6) and additional deer species included from GenBank…The full-length sequence alignment is shown in Supplementary Figure S1”
Line 90 “no pre or pandemic collected deer tested were positive for SARS-CoV-2 antibodies” It should be read “no pre or pandemic deer sera collected tested positive for SARS-CoV-2 antibodies”
Line 91 “with the exception of K-N substation in..” it should be read “substitution”
Line 105 “data on susceptibility of SARS-CoV-2 in European deer ” It should be read “data on susceptibility to SARS-CoV-2 in European deer”
Line 106 “receptor amongst cervids suggest similar susceptibility” It should be read “receptor residues amongst cervids suggests similar susceptibility”
Line 115 “the government independent of the land” It should be read “the government regardless of the land
Line 139 “contacts among central European deer populations may diminish..” It should be read “contacts among central European deer populations and may diminish..”
Supplementary Figure 1. “Full multiple sequence alignment of the ACE2 protein” It should be read “Multiple sequence alignment of the full-length ACE2 protein sequences..”
“The high variability in the aligned sequences of ACE2 from position 600 is delimited with a green line.” Which green line? The green line is missing in Supplementary Figure 2.
Moreover I suggest eliminating the Conclusion paragraph and merging it with the discussion section since the discussion without that part is inconclusive.
Author Response
There is a great deal of interest in reverse zoonotic transmission of SARS-CoV-2 and the manuscript by Moreira-Soto and colleagues provides additional information about the force of infection in wild animals in a COVID-endemic environment. The authors performed a serological screening on pre-pandemic and pandemic sera of 3 different deer species, using a previously validated pseudoneutralization assay method (Chandler et al), which allowed to confirm no prevalence of infection in Germany and Austria. Moreover, the authors performed an “in-silico” analysis to compare ACE2 receptor sequences in different animal species.
Although the aim of the work is overall clear, the text needs some minor language revision in order to make the reading of the text smoother, together with a better organization of presented data.
Response: We thank the reviewer for their positive assessment of our manuscript and helpful comments.
The sample size used by the authors is sufficient and adequate to infer results on the prevalence of infection on 3 different deer species although a main concern can be recognized in the lack of information about the geographical provenance of collected sera and the lockdown measures eventually adopted in those territories. Since Germany and Austria have an important deer population that is extending throughout the nations better geographical identification is mandatory, also considering that the 230 animals tested are a low percentage of the total deer population considered in comparison with the total numbers.
Response: The geographical provenance for the samples is provided in Supplemental Table 1. Please note that due to legal and privacy issues in Austria, the exact location (revier) is not provided. For the German samples this information is provided.
Here are some minor comments.
Line 38 pseudoneutralization assay instead of “assays”
Response: We have made the change as suggested
Line 46 “Sera was”.. sera were
Response: Corrected
Line 53 “equal volume of deer sera was pre-incubated ” instead of were
Response: Corrected
Line 56 “Absorbance values at 450 nm were” instead of readings
Response: Changed
Line 64 “All positive and negative controls were in the 450 nm range stated in the manufacturer’s instructions.” I suggest moving this statement in the results section
Response: moved
Figure 1 caption. “All positive and negative 69 controls were in the 450 nm range stated in the manufacturer’s instructions.” How can be distinguished from other samples? Are there in the image? Otherwise let this statement only in the M&M section. As well as “None of the samples approached the cutoff” can be written only in the result section.
Response: We removed the statement about the range so that is I in the results section per the previous comment. We moved the sample cutoff statement to the results.
“Results of the Pseudoneutralization point assay are shown by year and deer species. All positive and negative controls were in the 450 nm range stated in the manufacturer’s instructions. The assay cutoff for the cPass SARS-CoV-2 Neutralization Antibody Detection Kit is 30% inhibition which is is indicated. None of the samples approched the cutoff.”
It should be read “ Results of the Pseudoneutralization point assay are shown by year and deer species. Horizontal dashed line indicates the assay cutoff of the cPass SARS-CoV-2 Neutralization Antibody Detection Kit as 30% inhibition.”
Response:
Changed as suggested
Line 73 Check reference number. I suppose the reference the authors are referring to is the number 6 instead the number 5. The same for line 79, 85, and Supplementary figure caption.
Response: We thank the reviewer for catching this error and have fixed it throughout
Figure 2 caption. “Multiple sequences the ACE2 protein were performed with the species characterized in (5) and additional deer species included from GenBank. . The full alignment is in Supplementary Figure S1” It should be read “ Multiple sequence alignment of ACE2 protein of species characterized in (6) and additional deer species included from GenBank…The full-length sequence alignment is shown in Supplementary Figure S1”
Line 90 “no pre or pandemic collected deer tested were positive for SARS-CoV-2 antibodies” It should be read “no pre or pandemic deer sera collected tested positive for SARS-CoV-2 antibodies”
Response: Corrected
Line 91 “with the exception of K-N substation in..” it should be read “substitution”
Response: corrected
Line 105 “data on susceptibility of SARS-CoV-2 in European deer ” It should be read “data on susceptibility to SARS-CoV-2 in European deer”
Response: corrected
Line 106 “receptor amongst cervids suggest similar susceptibility” It should be read “receptor residues amongst cervids suggests similar susceptibility”
Response: Corrected
Line 115 “the government independent of the land” It should be read “the government regardless of the land
Response: Corrected
Line 139 “contacts among central European deer populations may diminish..” It should be read “contacts among central European deer populations and may diminish..”
Response: Corrected
Supplementary Figure 1. “Full multiple sequence alignment of the ACE2 protein” It should be read “Multiple sequence alignment of the full-length ACE2 protein sequences..”
Response: Corrected
“The high variability in the aligned sequences of ACE2 from position 600 is delimited with a green line.” Which green line? The green line is missing in Supplementary Figure 2.
Response: Corrected
Moreover I suggest eliminating the Conclusion paragraph and merging it with the discussion section since the discussion without that part is inconclusive.
Response: Changed as suggested
Reviewer 4 Report
The authors evaluated pre and pandemic exposure of German and Austrian deer species using a SARS-CoV-2 pseudoneutralization assay. This is an important study to assess the status of cervids common to Germany and Austria given that spillover of SARS-CoV-2 to North American white tailed deer has been well documented. Reporting the negative findings of no evidence of SARS-CoV-2 exposure in these European deer is important to add to the literature.
Given that hunters were involved in helping with sample collection, it would be useful to acknowledge guidance for hunters in North America and what could be applied in Europe to prevent many zoonoses that could be linked to deer. Additionally, the authors should acknowledge that this is a small number of deer over a 2-year time period and the importance of continuing to monitor for any changes in the European deer population as this can change over time.
Overall, this is a clear and well done study.
Author Response
The authors evaluated pre and pandemic exposure of German and Austrian deer species using a SARS-CoV-2 pseudoneutralization assay. This is an important study to assess the status of cervids common to Germany and Austria given that spillover of SARS-CoV-2 to North American white tailed deer has been well documented. Reporting the negative findings of no evidence of SARS-CoV-2 exposure in these European deer is important to add to the literature.
Given that hunters were involved in helping with sample collection, it would be useful to acknowledge guidance for hunters in North America and what could be applied in Europe to prevent many zoonoses that could be linked to deer. Additionally, the authors should acknowledge that this is a small number of deer over a 2-year time period and the importance of continuing to monitor for any changes in the European deer population as this can change over time.
Overall, this is a clear and well done study.
Response: We thank the reviewer for their suggestion. We want to avoid suggesting guidance to North American hunters because our sample is relatively small and while we have come up with what we think is a likely scenario for why there is such a drastic difference, it is not immediately clear how this would be implemented in North America. “De-urbanizing” deer over such a vast range would be exceptionally difficult and brings up both practical and ethical aspects of animal management that are well beyond the scope of this report.